# OpenReview forum: "Catastrophic Cyber Capabilities Benchmark (3CB): Robustly Evaluating LLM Agent Cyber Offense Capabilities"
_ICLR.cc/2025/Conference — Submitted to ICLR 2025_

### Official Review · Reviewer_CUpv · 2024-10-26

**Soundness:** 2
**Presentation:** 2
**Contribution:** 2
**Rating:** 5
**Confidence:** 4

**Summary:**

This paper introduces a new benchmark called 3CB, which includes 15 challenges used to evaluate the cyber offensive capabilities of different LLMs. This paper then conducts experiments on a set of agents with different LLMs and shows that larger LLMs tend to have stronger cyber offensive capabilities than smaller LLMs.

**Strengths:**

1. The motivation is reasonable and has real-world implications.

2. This is the first benchmark for evaluating the catastrophic cyber capabilities of LLMs.

**Weaknesses:**

Although the authors claim that this is the first comprehensive benchmark for evaluating the catastrophic cyber offensive capabilities of LLMs, the technical contribution is not strong enough to pass the high bar of ICLR. The presentation of the paper is more like a technical report, which is not as well-organized as a technical research paper. For experiments, only some superficial results are obtained, e.g., larger LLMs tend to have stronger cyber offensive capabilities than smaller LLMs, which makes the paper seem neither suitable for a technical paper nor a dataset/benchmark paper (in some other top conferences). Therefore, there is a quite large space for improvement before the paper is ready for acceptance.

**Questions:**

See Weaknesses.

---

> ### Author Response · Authors · 2024-11-25
>
> Thank you for your feedback.
>
> On the point that the paper is unsuitable as a benchmark, we respectfully disagree, as there are no other papers taking a systematic, literature-grounded approach to the challenge curation process – our incorporation of the ATT&CK categories. These categories are used in cybersecurity literature and practical applications and from what we can see, a structure for evaluation such as this is not currently represented in the literature. As an addendum, a recent literature review of cyber capabilities evaluation literature, whose authors we recently spoke with, identified the lack of systematic evaluation as a key limitation to the existing literature. After we spoke with them, they mentioned that we indeed captured one of their biggest points of critique.
>
> On experiments, we feel that the range of models and tasks used was wide, representing most state-of-the-art models, including open source models, as well as focusing on representativeness of the challenges as we have mentioned above. Could you share some experiments that you would be interesting for us to run?

---

> > ### Author Response · Authors · 2024-11-30
> >
> > Dear Reviewer,
> >
> > We would like to follow up and ask if you have had time to consider our rebuttal.
> >
> > Regards,
> > The Authors

---

### Official Review · Reviewer_mLRf · 2024-10-30

**Soundness:** 2
**Presentation:** 3
**Contribution:** 3
**Rating:** 8
**Confidence:** 4

**Summary:**

The paper proposes an evaluation framework of state of the art LLM agents in their capabilities to carry out offensive cybersecurity tasks. Based on the MITRE ATT&CK framework, they propose 15 cybersecurity challenges, representing the 15 attack categories (one from each category) of the ATT&CK framework, that they use to evaluate the capabilities of 14 LLM agents in completing successfully each challenge. Each challenge is attempted 10 times with the best elicitation variation. Then, a linear mixed-effects model is used to quantify each agent's performance. The experimental results show a variability in the rate of completion between LLM agents, the best being the larger ones (GPT-4 and Claude 3.5 Sonnet). They also reveal a variability due to the communication protocol used by the agent, notably, XML outperforming Markdown and XML. The overall conclusion of the paper is that the proposed 3CB framework shows that LLMs (mainly the large ones) are capable of sophisticated cyber operations and hence their use in cybersecurity offensive operations should be regulated.

**Strengths:**

On the positive side,
1) The paper is very well written and enjoyable to read.
2) The topic is timely and very relevant, as penetration testing is still a manual process heavily involving human agents who need to be trained on offensive tactics and requiring a high technical expertise. Such framework is very welcome for organizations who opt for an offensive security strategy to harden their system security.
3) The proposed framework is a good step towards leveraging the capabilities of state of the art LLM agents for cybersecurity offensive tasks.
4) The approach is comprehensive as it covers all the categories of cybersecurity attacks (according to MITRE) and is evaluated on the major LLM agents currently.

**Weaknesses:**

On the negative side,
1) In my opinion, the paper does not show clearly the capabilities of LLM agents in cybersecurity offensive operations. Apart from the (good) example in Appendix B.2, as a reader I couldn't gauge the capabilities of the LLMs in cybersecurity attacks.
2) For example, the proposed challenges in Table 1 come with brief descriptions that does not allow to assess how simple/sophisticated they are. Take the first challenge, nmap, to me it is a straightforward challenge. Then, when you check the rate of completion for the various LLMs (Figure 4), you notice that not all LLMs could complete it. In particular, binpwn, which is a more demanding challenge, has a better completion rate. I find this very strange. Also if you take the setuid, you notice a higher completion rate than nmap. This suggests that setuid is even simpler than nmap ! I find these results counterintuitive and do not help to gauge well the capabilities of the LLMs.
3) Apart from binpwn, no details are provided for the other challenges (I understand that 4 of them are kept confidential on purpose). I tried to get those details from the github repository, but as far as I saw, it contains just a demo of again the binpwn. Nothing on the other challenges. I may be wrong.
4) I couldn't run the code because the repository lacks clear instructions on how to install the software and run it.

So as recommendation, I would suggest providing more details about each challenge (in an extensive version of Table 1, or better yet, in the appendix, similair to binpwn). Also, list all the prompts used in the evaluation (in the appendix or in the github).

**Questions:**

1) How the prompts are chosen for each challenge? As I understand, several prompts were used for every challenge.
2) Some challenges require specific environment setup (VMs, specific network configuration, etc.). How those environment requirements are handled in the framework ?
3) Can you explain why some LLMs complete some sophisticated challenges but fail in simpler ones ? This is crucial to understand better the capabilities of the LLMs in cybersecurity offensive operations.

---

> ### Author Response · Authors · 2024-11-25
>
> A sincere thank you for engaging with our work so thoroughly. We address your comments on the weaknesses below:
>
> # On Weaknesses
>
> 1. For understanding the challenges in greater detail, please take a look at our run visualizer website:
> https://copper-autonomy-deteriorate.github.io/
> To see details of each run, click into each run (i.e. Run 15388). You may have to wait for some time for the results to load. Afterwards, the runs are visible in the pane that appears to the side.
> To address your point on the disparity in the perceived difficulty of the challenges, you make a great point about how what is easy to a human may not be easy to an agent.
> Although nmap is an easy tool for a human to use, it is brittle in that the challenge requires adherence to the communication protocol to invoke the tool and parse its outputs successfully, which is a big hurdle for smaller models. Please see the answer to question 3 for more discussion.
>
> 2. Concretely, each run is fully specified by a configuration file that contains the docker container(s) needed, setup files, starting prompt for the LM agent, etc. For instance, for the nmap challenge, the spec may be found here:
> https://github.com/copper-autonomy-deteriorate/outline-chord-transmission/blob/main/task_configs/nmap.toml
> Additionally, the full specifications of each challenge are in this folder of our Github Repository:
> https://github.com/copper-autonomy-deteriorate/outline-chord-transmission/blob/main/task_configs/
> This is with the exception of our 4 excluded challenges: dfbit, bashhistory, sshhijack, cnc.
>
> 3. We have updated the README with more detailed instructions of how to run some sample challenges. [(link)](https://github.com/copper-autonomy-deteriorate/outline-chord-transmission/blob/main/README.md)
>
> # On Questions
>
> 1. For every challenge, together with the definition itself (environment, goal, max number of turns, etc), a range of prompts is created manually. This set of prompts is intended to cover a range of conceptual framings (offensive agent, security competition, abstract problem) and communication protocols (as some models are very sensitive to formatting). For some challenges, the prompts also include hints, in order to extract a more meaningful signal from run logs. All in all, coming up with the set of different prompts+protocols is intended to represent a good-faith attempt at crossing the elicitation gap of prompt engineering. Future work could include automated prompt optimization a-la https://arxiv.org/pdf/2305.03495.
> 2. Specific environment requirements are encoded in the challenge definition TOML files. A challenge is assumed to need 1 or more docker containers joined in a network, with each container having their own files (including the Dockerfile from which the image is built), and container-level metadata, such as its network IP address or the necessary RAM size.
> 3. We think there’s several non-trivial factors at play here:
>     1. The question of definitions of “success” and “fail”. On the level of individual runs, stochasticity of LLMs has significant influence. While a weaker model is unlikely to solve a hard challenge end-to-end by rolling a lucky token, a stronger model can definitely go into unhelpful rabbit holes. The bashhist challenge is basically a sanity check, with the answer hidden in a sea of red herrings, necessitating a degree of perspective from the model. Still, from the logs we can see that many strong models get misled and run out of tokens.
>     1. In our view, for a given model and a challenge, the most meaningful problem framing is “Can the challenge be solved at all even once, given strong elicitation and many runs?”. It is only bottlenecks on challenge creation and token budgets that force us to rely on measures like Average Success Rate, which is more likely to exhibit the behavior you point out.
>     1. But most importantly, what is easy or hard for a human cybersecurity practitioner is not necessarily the same for LLMs. While there’s a slew of trivial failings that some LLMs display (failure to follow the protocol, messing up the environment state, not being able to use TUI elements like “press c to continue”, etc.), the issue is much deeper. The text format, communication protocols, turn limitations, and inherent differences between how humans and LLMs interact with the environment make direct comparisons of challenge difficulty hard. We conjecture that any motivated human practitioner will be able to solve all challenges in our set, given time and tools, but even the strongest model will struggle to solve all of them, all the time, in the message budget allotted.

---

> > ### Comment · Reviewer_mLRf · 2024-11-26
> > **Checked the visualizer**
> >
> > Thanks for the clarifications. While I checked the github repository, I didn't check the visualizer. It was not clear in the reproducibility statement. Now I checked the visualzer and could assess better the proposed system.
> > However, I am still not convinced about the disparity in the performance between "easy" and "challenging" tasks. Providing hints, which I now see through the visualizer, is one possible explanation. Unfortunately, hints are not discussed in the paper.

---

### Official Review · Reviewer_hgnu · 2024-10-31

**Soundness:** 3
**Presentation:** 3
**Contribution:** 1
**Rating:** 3
**Confidence:** 4

**Summary:**

This paper proposes a framework and benchmark of 15 challenges that allows a LLM to conduct attacks. They evaluate a number of LLMs in this framework.

**Strengths:**

This is a benchmarks paper and I believe the authors have chosen a primary area that is not suited for this paper. If I evaluate this according to the choice of authors' choice of primary area, then this paper should be rejected as there is no new technique in safety, alignment, privacy, or social considerations.

Merits:
1) The problem is interesting and a good direction to study and benchmark the behavior of LLMs.
2) The experiments have been done extensively (but as acknowledged by authors, not covering all tactics)
3) This can be a valuable basis for further engineering.

**Weaknesses:**

Weakness:
1) This line "While some benchmarks claim to measure general cyber capabilities but only cover specific sub-capabilities, 3CB ensures that each challenge is designed such that successful completion by a model accurately reflects its ability to apply the technique described in that challenge." First, this line requires citations to the "some" benchmarks and then I did not understand how 3CB ensures that each challenge is designed such that successful completion by a model accurately reflects its ability to apply the technique described in that challenge? Where is the evidence for this claim?
2) Also, the authors claim novelty of the challenges - how is this ensured. Particularly, when the challenges are broadly in the known ATT&CK category? Also, LLMs can easily relate similar (but not same) textual concepts, so where is the this novelty coming from - are these conceptually new challenges?
3) Related to above, the authors say 11 challenges are publicly released - does this mean on the internet?
4) There are many open questions arising from the work, some of which should be done to make the work stronger? Will a cybersecurity-specific finetuned smaller LLM (like Llamma) do better? Clearly, large LLM like GPT have many advantages in terms of training data and size of model, more fine-grained comparison of why these succeed and smaller LLM fail is needed.

Overall, I find this to be useful work, but I am not sure if it fits well in a top research conference.

**Questions:**

Please respond to the weaknesses.

---

> ### Author Response · Authors · 2024-11-25
>
> Dear reviewer,
>
> We appreciate your positive perspective on our work's future potential and our experiments. Regarding your concerns:
>
> You mention that our paper does not introduce new approaches within the topic of “alignment, fairness, safety, privacy, and societal considerations”. We respectfully disagree, as there are no other papers taking a systematic, literature-grounded approach to the challenge curation process – our incorporation of the ATT&CK categories. These categories are used in cybersecurity literature and practical applications and from what we can see, a structure for evaluation such as this is not currently represented in the literature. As an addendum, a recent literature review of cyber capabilities evaluation literature, whose authors we recently spoke with, identified the lack of systematic evaluation as a key limitation to the existing literature. After we spoke with them, they mentioned that we indeed captured one of their biggest points of critique.
>
> ## Weaknesses
>
> 1. We reference these benchmarks in the Related Works section, but you are correct that we can make it clearer in the methodology section as well. The evidence for this claim is in the scaffolding we have built. Whereas existing benchmarks often use a question-answering structure (e.g. the [WMDP benchmark](https://www.wmdp.ai/)), we use a realistic simulation of a network of multiple computers that the AI agent interacts with. Whereas we cannot be confident that an agent who answers a question about extracting passwords from the bash history correctly can actually extract said information in a real-world setting, I can be confident that the same agent can extract passwords from the bash history if they're able to do it in a realistic multi-step interactive simulation, such as what the 3cb harness provides.
>
> 2. We do not construct the challenges from existing material. Whereas similar work copy existing challenges from the web (see e.g. [SWE-bench](https://www.swebench.com/), [CyBench](https://cybench.github.io/), and others), we create our own original challenges guided by the spirit of the tactic categories. I.e., they are novel in design and avoid being in the training dataset of any models.
>
> 3. Yes, the challenges are publicly released on both Github: https://github.com/copper-autonomy-deteriorate/outline-chord-transmission/ and in an interactive format (demonstration) on our website https://copper-autonomy-deteriorate.github.io/.
>
> 4. There’s a long list of next steps one can take from our work in this paper. However, we will argue that we adequately cover the novelty of introducing a proper systematic design to the challenge creation process and that the validation of said approach is represented in the paper through the results we present. While it would be very interesting to dive deeper into the results, our further explorations (e.g. clustering of the challenge runs and visualization of these, manual investigations of the runs, etc.) are not included due to the page limit. Additionally, it's not crucial to the arguments we present. To expand on the next steps you propose:
>
> *Can specialized LLMs/Agents do better than e.g. vanilla gpt-4o?*
> They probably would! However, what we also show is that despite the differences in training approaches for the models (some more on code than others), the best ones are still the largest ones (GPT-4o and Claude).
>
> *Why do some models succeed and some don't?*
> We include analyses of the elicitation gaps in our research to try to extract the most capable versions of each model. As mentioned in the paper, the best-performing combinations for each model is included in the results and we have these results, though we prioritized other parts of our experiments in the main paper.

---

> > ### Author Response · Authors · 2024-11-30
> >
> > Dear Reviewer,
> >
> > We would like to follow up and ask if you have had time to consider our rebuttal.
> >
> > Regards,
> > The Authors

---

> > > ### Comment · Reviewer_hgnu · 2024-12-02
> > > **Concerns remain**
> > >
> > > I have looked at the rebuttal, actually long ago but did not know what to say. The issue here is a matter of perception, and my perception has not really changed. My main concern being that I do not see much advance in ML or AI here. I am not an expert in cybersecurity, but I am not getting the importance here.

---

> ### Author Response · Authors · 2024-12-02
>
> Dear Reviewer,
>
> Thank you for your continued engagement with our work.
>
> > I do not see much advance in ML or AI here
>
> Benchmarks, by design, do not typically advance ML/AI capabilities directly, giving "no new technique in safety, alignment, privacy, or social considerations". Rather, they serve the crucial scientific function of measuring and understanding current capabilities. Just as GLUE and SuperGLUE didn't advance language model capabilities directly but provided essential insights into model performance, our benchmark provides systematic understanding of where current AI systems stand regarding multi-step reasoning in practical scenarios about *highly relevant and impactful cyber-offense scenarios*, and should be judged on this basis.
>
> Our benchmark makes several key contributions to ML/AI research by:
>
> 1. Providing the first systematic evaluation framework for complex, multi-step reasoning in interactive scenarios with our outlined taxonomy, providing a roadmap to comprehensive coverage of cyber attacks by AIs.
> 2. Revealing important insights about current model capabilities and limitations
> 3. Establishing reproducible metrics for tracking progress in this domain
>
> We would welcome specific suggestions for better communicating these contributions in the paper.
>
> Best regards,
> The Authors

---

### Author Response · Authors · 2024-11-25

Dear Reviewers,

We sincerely thank you for your thoughtful and constructive feedback on our paper. Your insights have been invaluable in helping us improve our work. We have carefully considered all of your comments. Once again, we thank all reviewers for their valuable time and insights.

If you have not viewed our results and the challenges themselves yet, please take a look at our Run Visualizer Website and Github Repository

**RUN VISUALIZER** https://copper-autonomy-deteriorate.github.io/

**GITHUB REPO** https://github.com/copper-autonomy-deteriorate/outline-chord-transmission/

We further address your points in the individual comments below.

Sincerely, The Authors

---

### Meta-Review · Area_Chair_FQBD · 2024-12-19

**Metareview:**

While this work presents a wide range of benchmark tasks for the domain of cybersecurity, it is not clear, in its current state, how these evaluations provide information that is not currently available in other benchmarks, such as CyberSecEval 2. The authors mention these other benchmarks in the related works, but do not detail how their work relates to these predecessors. Importantly, many of the tasks in the C3B evaluationa are nearly fully solved by existing models, such as GPT-4o and Sonnet 3.5, and Llama 3.1. Indeed, only around 6 tasks in the benchmark seem to present challenges to current generation LLMs. Moreover, there is a notable lack of deeper analysis on where models fail on these tasks (ie what exactly makes them challenging?) as well as details on the elicitation strategies used per task. I recommend the authors resubmit the improved paper to a venue such as the NeurIPS Datasets and Benchmarks track.

**Additional Comments On Reviewer Discussion:**

The reviewers largely agree the scientific contribution of this paper runs shallow, as the proposed benchmark does not provide sufficient details around the specific task and their associated challenges.

---

### Decision · Program_Chairs · 2025-01-22

Reject